# Screening of Anisakis-Related Allergies and Associated Factors in a Mediterranean Community Characterized by High Seafood Consumption

**DOI:** 10.3390/foods13172821

**Published:** 2024-09-05

**Authors:** Santo Fruscione, Maria Barrale, Maurizio Zarcone, Davide Alba, Barbara Ravazzolo, Miriam Belluzzo, Rosa Onida, Gaetano Cammilleri, Antonella Costa, Vincenzo Ferrantelli, Alessandra Savatteri, Daniele Domenico De Bella, Salvatore Pipitone, Alida D’Atria, Alessia Pieri, Fabio Tramuto, Claudio Costantino, Carmelo Massimo Maida, Giorgio Graziano, Marialetizia Palomba, Simonetta Mattiucci, Ignazio Brusca, Walter Mazzucco

**Affiliations:** 1PROMISE Department, University of Palermo, 90127 Palermo, Italy; davide.alba@unipa.it (D.A.); miriam.belluzzo@unipa.it (M.B.); alessandra.savatteri@unipa.it (A.S.); danieledomenico.debella@unipa.it (D.D.D.B.); fabio.tramuto@unipa.it (F.T.); claudio.costantino01@unipa.it (C.C.); carmelo.maida@unipa.it (C.M.M.); walter.mazzucco@unipa.it (W.M.); 2U.O.C. of Clinical Pathology Buccheri La Ferla Hospital FBF, 90123 Palermo, Italy; marbarrale@gmail.com (M.B.); rossellaonida@gmail.com (R.O.); brusca.ignazio@fbfpa.it (I.B.); 3U.O.C. Epidemiologia Clinica con Registro Tumori, Azienda Ospedaliera Universitaria Policlinico di Palermo, 90127 Palermo, Italy; maurizio.zarcone@policlinico.pa.it (M.Z.); barbara.ravazzolo@policlinico.pa.it (B.R.); salvopipi@gmail.com (S.P.); allidatria@gmail.com (A.D.); alessia.pieri@policlinico.pa.it (A.P.); giorgio.graziano@unipa.it (G.G.); 4Experimental Zooprophylactic Institute of Sicily, 90129 Palermo, Italy; gaetano.cammilleri86@gmail.com (G.C.); antonella.costa@izssicilia.it (A.C.); vincenzo.ferrantelli@izssicilia.it (V.F.); 5Department of Ecological and Biological Sciences, Tuscia University, 01100 Viterbo, Italy; m.palomba@unitus.it; 6Section of Parasitology, Department of Public Health and Infectious Diseases, University Hospital “Policlinico Umberto I”, Sapienza-University of Rome, 00185 Rome, Italy; simonetta.mattiucci@uniroma1.it

**Keywords:** *Anisakis* allergy, *Anisakis* IgE sensitization, basophil activation test

## Abstract

Dietary changes expose consumers to risks from *Anisakis* larvae in seafood, leading to parasitic diseases and allergies. *Anisakis* is recognized by EFSA as a significant hazard, with potential oncogenic implications. Diagnostic advancements, like the Basophil Activation Test (BAT), enhance sensitivity and accuracy in identifying *Anisakis* sensitization, complementing traditional IgE tests. We conducted a cross-sectional study on patients with allergic symptoms from April 2021 to April 2023 at two outpatient clinics in western Sicily. Our goal was to assess the prevalence of *Anisakis*-related allergies and to identify risk profiles using specific *Anisakis* IgE and the BAT, especially in regions with high raw fish consumption. The study evaluated specific *Anisakis* IgE as a screening tool for *Anisakis* sensitization, using questionnaires, blood samples, and immuno-allergology analyses. *Anisakis*-specific IgE values were compared with the BAT results, with statistical analyses including Fisher’s exact test and logistic regression. The results showed an 18.5% seroprevalence of *Anisakis* IgE, while the BAT as a second-level test showed 4.63%, indicating the BAT’s superior specificity and accuracy. The study highlighted the importance of the BAT in diagnosing *Anisakis* sensitization, especially in cases of cross-reactivity with Ascaris and tropomyosin. The findings confirm the BAT’s exceptional specificity in identifying *Anisakis* sensitization and support using *Anisakis*-specific IgE for population-based risk profiling. The BAT can effectively serve as a confirmatory test.

## 1. Introduction

Changes in eating habits have made consumers susceptible to potential dangers associated with parasitic diseases and allergies caused by the presence of *Anisakis* larvae parasite in the consumed seafood [1,2]. *Anisakis*, the agent responsible for these health hazards, can initiate anisakiasis, an infectious disease marked by gastrointestinal symptoms and/or diverse allergic reactions [3]. Because of the severity of these conditions, the European Food Safety Agency has recognized *Anisakis* among the most relevant biological hazards in seafood [4]. The culprits behind this ailment, *Anisakis pegreffii* and *Anisakis simplex*, not only serve as agents provoking gastric and intestinal anisakiasis, but also elicit allergic reactions in individuals who have become sensitized [3]. Thus, it has been hypothesized that in some Mediterranean areas characterized by high raw, marinated, or smoked fish consumption, there could be an excess risk of exposure to *Anisakis*, together with a surge in allergic reactions of unknown food origins, or those erroneously attributed to other causes, that might be explained by sensitization to this parasite [5,6].

Due to the increasing attention paid to this phenomenon, several studies have been conducted with the aim of improving diagnostic techniques [7,8,9]. A recent study confirmed the Basophil Activation Test (BAT) as the most specific (100%) and accurate (92.45%) diagnostic test for *Anisakis* sensitization, with an acceptable level of sensitivity (84.62%), as compared with the dosage of IgE against *Anisakis*, which had a higher sensitivity (92.31%) but a lower accuracy (64.15%) and a significantly lower specificity (37.04%) (9). Therefore, although the *Anisakis*-specific IgE test, due both to its low cost and ease feasibility together with its high sensitivity, has been used in population-based seroprevalence investigations, its low specificity could have led to an overestimation of *Anisakis* sensitization [7]. Thus, the use of the BAT as a second-level diagnostic test has been recommended to confirm positivity to *Anisakis*-specific IgE [9]. Historically, the cutoff level of specific IgE for allergy diagnosis has been set at 0.35 kUA/L, a value derived from second-generation analytical platforms and considered close to the threshold of clinical manifestations; meanwhile, with the availability of third-generation tests, the analytical sensitivity target has dropped to 0.10 kUA/L [10].

We aimed to assess whether the analytical cut-off for *Anisakis*-specific IgE can be used as a screening test for *Anisakis* susceptibility as an alternative to the clinical cut-off level. To this end, we conducted an observational study on a sample of ambulatory outpatients from western Sicily, a Mediterranean island characterized by a significant consumption of raw, smoked, or marinated fish. Factors potentially associated with *Anisakis*-related allergies were also investigated.

## 2. Materials and Methods

A cross-sectional observational study was carried out in a group of outpatients experiencing allergic manifestations, who had consecutive access, between May 2022 and April 2024, to the allergology outpatient ambulatories of the Regional Reference Centre for Immunoallergology of the “Buccheri La Ferla Fatebenefratelli” Hospital in Palermo and of the “San Giovanni di Dio” Hospital of the Agrigento Provincial Health Agency, both located in western Sicily, South Italy.

Inclusion criteria were a history suggestive of sensitization to *Anisakis* in subjects reporting clinical allergic manifestations (asthma, rhinitis, conjunctivitis, urticaria and/or angioedema, abdominal pain, diarrhea, vomiting or anaphylaxis) after ingesting fish. We also included outpatients with symptoms of urticaria lasting more than six weeks as they were potentially affected by a chronic *Anisakis*-related allergy. The exclusion criterion was a sensitization to fish documented by diagnostic tests.

Upon recruitment, every participant received and subscribed an informed consent form outlining the study’s details. Subsequently, a digitally formatted questionnaire was administered, designed to systematically gather socio-demographic details, dietary habits, contact with raw, smoked, or marinated fish, and other pertinent information related to potential exposure to *Anisakis*. Information was collected anonymously, and each subject was associated with an alphanumeric identification code to ensure privacy. A 10 mL blood sample was collected from each outpatient to perform the immuno-allergology analyses. To this end, samples were centrifugated and sera were aliquoted and stored at −20 °C. Outpatients negative to fish allergens and positive to *Anisakis* extracts were tested with specific IgE for Ascaris (p1) and crayfish tropomyosins (f351) and with mite-specific IgE (*Dermatophagoides pteronyssinus*, d1). The specific IgE sensitization was tested by using the Immunocap method [11]. A further 10 mL blood sample was taken to perform the BAT, when appropriate. The BAT (positivity ≥ 15%) was performed using a flow cast kit (Bühlmann Laboratories AG, Schönenbuch, Switzerland) and home-made *Anisakis* extracts, obtained from *Anisakis pegreffii* (A.pe.) and used at a concentration of 22.5 ng/mL. Each sample was associated with an alphanumeric code relating to the questionnaire administered to each outpatient.

*Anisakis*-specific IgE values detected by the clinical cut-off (positivity > 0.35 kUA/L) and analytical cut-off (positivity > 0.10 kUA/L) were used to calculate the prevalence of *Anisakis* sensitization and were, then, compared to the BAT results.

The study was conducted in accordance with the Declaration of Helsinki, and the protocol was approved by the Ethics Committee of Palermo 1 (n. 107, 8/2018).

### Statistical Analysis

Absolute and relative frequencies (percentages) were considered in the descriptive analysis. Fisher’s exact test was performed to compare outpatient groups for qualitative variables, while permutation tests were used for quantitative variables. A comparison by Fisher’s exact test was made between the different patient groups, detecting both *Anisakis*-specific IgE levels by clinical cut-off and analytical cut-off, and the BAT values.

Univariate analyses were performed, and Odd Ratios (ORs) were calculated with their 95% confidence intervals (CI95%), following Blaker’s procedure [12]. A Haldane–Anscombe correction was implemented when at least one cell in the 2 × 2 contingency tables exhibited a frequency of 0.

Multiple logistic analyses were conducted to obtain the adjusted ORs, by means of a stepwise regression procedure, backward, following the Akaike Information Criterion (AIC) minimization. The variables included in the model were selected based on their statistical significance in univariate analyses and their clinical relevance. The final multiple logistic model was evaluated for goodness-of-fit using the Hosmer–Lemeshow test, and multicollinearity was assessed using the generalized Variance Inflation Factor (VIF) [13,14].

Statistical significance was set at *p*-value less than 0.05.

## 3. Results

Table 1 shows the distribution of the socio-demographic characteristics of the sample of 259 participants recruited in the study.

Females were prevalent (n. 171, 66.0%) in our sample. The mean age of the recruited patients was 40.8 years old (SD = 20.5), with no statistically significant difference reported between males and female (*p*-value: 0.25). Regarding the education level, 36.7% (n. 95) of the sample reported having a secondary school diploma, 36.3% (n. 94) a middle school diploma, 12.4% (n. 32) a primary school licence, 10.4% (n. 27) a university degree, and 4.2% (n. 11) of the sample did not have any educational qualification. Anyway, no difference was highlighted by these groups (*p*-value: 0.87).

Concerning the area of residence, 51% (n. 132) of the recruited individuals resided in coastal urban areas, 30.9% (n. 80) in rural areas with direct access to the sea, and 18.1% (n. 47) in inland municipalities, but no statistically significant difference was reported between the three groups (*p*-value: >0.05). Lastly, 90% of the individuals (n. 233) had resided in their current location for more than 10 years, whereas the remaining 10% of the study sample (n. 26) had a residence duration of less than 10 years, but no statistically significant difference was highlighted (*p*-value: 0.09).

In Appendix A more findings obtained from the responses to the questionnaire are reported, covering information on dietary habits, contact with raw, smoked, or marinated fish, and other details related to the potential exposure to *Anisakis*. In the whole study sample, when using the analytical cut-off (>0.1 kUA/L), the IgE *Anisakis* specific positivity was 18.5% (n. 48); meanwhile, when using the clinical cut-off (>0.35 kUA/L), the seroprevalence detected by *Anisakis* specific IgE resulted in 7.3% (n. 19) (Table 2).

However, the prevalence decreased to 4.6% (n. 12/259) when using the BAT second level test (Table 2). Of interest, the BAT allowed us to highlight the presence of two false negative cases of *Anisakis* sensitization when using the IgE clinical cut-off, which instead were detected using the analytical cut-off (Table 2). More in depth, when considering the analytical cut-off, of the 48 subjects that tested positive for *Anisakis*-specific IgE, 12 were positive and 36 were negative at the BAT (*p*-value < 0.001); on the other hand, no subject negative for specific *Anisakis* IgE had a positive BAT result, confirming that this second-level test was highly specific in the diagnosis of *Anisakis* sensitization.

Table 3 reports the univariate comparison between variables obtained by the questionnaire administration and the laboratory results (*Anisakis*-specific IgE positivity by analytical cut-off and BAT positivity).

Outpatients positive to *Anisakis*-specific IgE had a higher mean age as compared to those negative (46.6 vs. 39.4 years old) (*p*-value = 0.03), and these results were confirmed by the BAT (54.3 years old for positives versus 40.1 years old for negatives; *p*-value = 0.02). Female outpatients had significantly lower odds of *Anisakis* IgE positivity as compared to males (OR: 0.32 [95%CI: 0.17; 0.61]; *p*-value < 0.001), but this excess of risk was not confirmed when using the BAT (OR: 0.35 [95%CI: 0.10; 1.17]; *p*-value < 0.08) (Table 3). The consumption of raw, marinated, or smoked fish was statistically significant inversely associated with *Anisakis* IgE positivity (OR = 0.14 [95%CI: 0.04; 0.49]; *p*-value = 0.002) and with BAT positivity (OR = 0.11 [95%CI: 0.03; 0.61]; *p*-value = 0.01), respectively (Table 3). Skin contacts with fish also resulted in a statistically significant inverse association with *Anisakis* IgE positivity (OR = 0.16 [95%CI: 0.09; 0.72]; *p*-value = 0.011) and BAT positivity (OR = 0.20 [95%CI: 0.05; 0.99]; *p*-value = 0.044), respectively (Table 3). Moreover, as expected, an extremely high significant association was found between *Anisakis* IgE positivity and BAT positivity (O.R.: 144.9 [95%CI: 17.2; ∞]; *p*-value < 0.001) (Table 4).

In the same direction, statistically significant (*p*-value = <0.001) estimates on the association between *Anisakis*-specific IgE positivity or BAT positivity and positivity to Ascaris-specific IgE and tropomyosin-specific IgE were also found (Table 4). The multiple logistic model, considering *Anisakis*-specific IgE positivity (≥0.1 kUA/L) as the dependent variable, highlighted a statistically significant association between *Anisakis*-specific IgE and the following variables: female sex, which resulted a potential protection on *Anisakis* sensitization (OR = 0.04 [95%CI: 0.01–0.27]; *p*-value = 0.001); age, resulting in a 7% excess risk per each year of increase in age (OR = 1.07 [95%CI: 1.02–1.11]; *p*-value = 0.002) (Table 5).

The multiple logistic model (Table 6), considering BAT positivity (≥15%) as the dependent variable, highlighted that every one year of increasing in age was associated with a 9% higher probability of having BAT levels ≥ 15% (OR = 1.09 [95%CI: 1.02–1.16]; *p*-value = 0.008), while the presence of angioedema was inversely associated with a BAT positivity (OR = 0.01 [95%CI: 0.00–0.69]; *p*-value = 0.03); the final model was well-fitted, with a Hosmer and Lemeshow goodness of fit test *p*-value of 0.941. The generalized variance-inflation factors ranged from 1.16 to 3.81, indicating low multicollinearity.

Figure 1 shows a flow chart translating a summary of the previous results into practical implications for the diagnosis and the screening of *Anisakis* allergy.

## 4. Discussion

We performed a cross-sectional observational study on a sample of 259 patients suffering from allergic manifestations in an epidemiological context characterized by a high consumption of raw, smoked, and marinated fish. More in depth, we tested specific *Anisakis* IgE as a screening test for *Anisakis* sensitization using the analytical cut-off as compared with the use of the clinical cut-off. Then, we used the BAT as a confirmation test for *Anisakis* sensitization.

In our study sample, the seroprevalence of specific *Anisakis* IgE was equal to 7.3% when using the clinical cut-off, and it raised to 18.5% when using the analytical cut-off. However, the prevalence decreased to 4.63% when using the BAT as a second-level confirmatory test. This finding aligns with recent studies, indicating that the BAT exhibited higher specificity, accuracy, and predictive value, as compared to *Anisakis*-specific IgE, while *Anisakis*-specific IgE demonstrated a better sensitivity than the BAT, while suffering low specificity at the same time. Of interest, the first-line screening conducted in our sample by using the *Anisakis*-specific IgE analytical cut-off allowed us to identify two false negative cases of *Anisakis* sensitization, confirmed to be positive by the BAT [9]. These two cases were retrospectively investigated and were both confirmed to have a previous positivity to *Anisakis*-specific IgE, tested using the clinical cut-off level, with a history suggestive of *Anisakis* allergy and being positive by the BAT.

In this scenario, the importance of using a highly sensitive test for diagnosing diseases induced by the *Anisakis* nematode is evident. The BAT appears to be particularly relevant as it allows for the verification of the clinical relevance of sensitization. This is crucial not only for diagnosing allergies, but also for monitoring other more serious complications, such as the possibility of larvae mimicking tumour-like masses or the infestation itself being a risk factor for tumour development [15,16,17,18].

In a 2018 systematic literature review exploring the prevalence of *Anisakis* sensitization across different study samples and diagnostic tests, it was revealed that estimates of *Anisakis* hypersensitivity varied significantly [7]. This variability was attributed to factors such as geographical location, population characteristics, diagnostic criteria, and the use of laboratory tests with differing sensitivity and specificity. Additionally, the findings of the systematic review led to the conclusion that *Anisakis* may be a concealed contributor to numerous adverse reactions, following the consumption of undercooked or raw fish, often mischaracterized as various forms of “fish allergy” [7].

*Anisakis* sensitization in the asymptomatic general population was identified in a range from 0.4% to 27.4% using indirect ELISA or by detecting *Anisakis*-specific IgE through Immunocap [11,19]. The prevalence varied between 6.6% and 19.6% when assessed by the skin prick test (SPT) [11,20,21,22]. Considering anamnestic criteria (symptoms after fish consumption), *Anisakis* allergy was found in 0.0% to 14.0% of patients [5,23,24]. Sensitization rates in samples selected from hospitalized subjects, depending on the criteria used to define allergy to *Anisakis*, ranged from 0.5% to 20% when IgE was >0.35 kU/L [25,26]. To this end, both indirect ELISA and Immunocap techniques were used. As mentioned, higher rates of hypersensitivity were observed in selected samples of symptomatic, allergic individuals who regularly consumed raw or undercooked fish, aligning with the established association between *Anisakis* sensitization, urticaria/allergic symptoms, and the consumption of undercooked fish [27,28]. Conversely, prevalence rates tended to be lower in studies with larger sample sizes, when diagnostic techniques focused on fewer but more specific *Anisakis* antigens, or when a higher threshold of positivity was applied for detecting specific antibodies [29].

In line with previous evidence, in our sample the univariate analyses highlighted an inverse association between fish intake or skin contact and positivity to both AS-specific IgE or the BAT, therefore suggesting that the development of an *Anisakis* allergy in the past may have led to a reduction in raw, marinated, or smoked fish consumption or contact [8,30]. Prevalence rates were markedly influenced by the *Anisakis* antigens selected as targets for the diagnostic tests, revealing substantial differences between crude extracts of whole *Anisakis* larvae and specific recombinant excretory proteins. Crude extracts of *Anisakis* larvae may encompass multiple allergens with cross-reactivity to other nematodes, crustaceans, insects, or mites [31,32,33,34,35]. Their use as target antigens in commercial tests, both serological (ImmunoCAP) and clinical (SPT), may lead to a reduced specificity and a subsequent overestimation of seroprevalence. In Croatia, the ELISA method employed for detecting rAni s1 and rAni s7 antigens on a sample of 500 healthy subjects from different areas revealed a prevalence decrease from a maximum of 3.5%, documented among individuals living on islands (presumed to be high consumers of fish), to 1.5% in urban coastal areas, while a prevalence of 0.0% was reported in the rural part of the country (an area of low fish consumption), emphasizing the association between *Anisakis* sensitization and fish consumption [36].

The BAT, utilizing flow cytometry, was introduced by Gonzalez-Munoz et al. in 2005 [37]. In a study by Frezzolini et al. comparing the BAT with the SPT and ImmunoCAP for diagnosing *Anisakis* sensitization among patients with chronic urticaria, atopic subjects, and healthy controls, all three tests demonstrated a good sensitivity [20]. However, the BAT achieved the highest specificity, reaching 100%, leading to the conclusion that the BAT is the most reliable diagnostic tool for anisakis allergy diagnosis [20].

In light of the aforementioned findings, it was deemed beneficial to explore potential associations among the available variables with both *Anisakis*-specific IgE positivity and BAT positivity, whereas in our study, *Anisakis*-specific IgE was initially employed followed by the BAT. Subsequently, the univariate analyses unveiled statistically significant associations between the positivity identified by both methods and the variables “age”, “Ascaris IgE positivity”, and “Tropomyosin IgE positivity”. Differently than the BAT positivity, *Anisakis*-specific IgE positivity exhibited a significant association with female sex, indicating it as a protective factor. Of interest, the multivariable analysis allowed us to highlight how increasing in age was associated with a growing probability to develop a sensitization to *Anisakis* detectable both by specific IgE and the BAT. Moreover, females were found to have a lower risk of sensitization to *Anisakis* than males when using *Anisakis*-specific IgE. This evidence, taken altogether, suggests potential information for use in addressing the risk profiling of the target population to be screened for *Anisakis* sensitization; however, these findings need to be confirmed by further studies with larger sample populations. Furthermore, in contrast to the assessment of *Anisakis*-specific IgE positivity, BAT positivity was linked to both skin contact with and consumption of raw, marinated, or smoked fish. More precisely, it was observed that individuals testing positive with the BAT tended to consume raw/marinated fish less frequently, suggesting prior sensitization to *Anisakis*. Similarly, BAT-positive subjects were found to have less frequent skin contact with raw/marinated fish, indicating pre-existing sensitization to *Anisakis* as well.

Of interest, in our series of outpatients, angioedema was inversely associated with BAT positivity. Usually, the presence of an isolated angioedema does not have an allergic origin, while when associated with urticaria it can follow an allergic origin or, more frequently, angioedema is due to a chronic spontaneous urticaria.

Furthermore, in subjects with minimal fish consumption or those who had excluded fish products from their diet who were still testing positive for *Anisakis*-specific IgE, it is plausible to hypothesize that this positivity may stem from both fish allergy and an allergy to *Anisakis* proteins. Moreover, the multivariate model revealed that Ascaris-specific IgE and tropomyosin-specific IgE sensitization were no longer associated with BAT positivity.

Lastly, the difference between the multivariable models using *Anisakis*-specific IgE and the BAT should be interpreted by firstly considering that specific IgE dosage is an immunochemical test, while the BAT is a functional test. More in depth, the IgE dosage measures the presence of the analyte, while the BAT assesses the functional effects that the presence of IgE can determine.

Fish is strongly affected by the environment in which it lives and is therefore susceptible to be attacked by parasites. Some may pose a health risk to consumers if the fish is consumed raw or not fully cooked. In contrast, cooking or freezing is an effective way to ensure its safety. The presence of these parasites, in particular *Anisakis*, can be considered a natural condition and not a sign of alteration itself. It is indeed possible that dead *Anisakis* larvae may contaminate raw food, leading to sensitization, but the procedures of cooking or freezing are considered to reduce the risk. As explained in the literature, the oral intake of seafood containing allergenic *Anisakis* proteins from dead larvae rarely results in sufficient blood allergen content to induce systemic symptoms of *Anisakis* allergy [38]. These findings indicate that it is generally unnecessary to uniformly eliminate cooked or processed seafood from the diets of most patients. However, it is important to recognize that the ingestion of dead *Anisakis* larvae, even in cooked or processed forms, can still trigger allergic reactions in patients who are highly sensitive. Therefore, while managing dietary restrictions, it is essential to tailor recommendations to individual sensitivities and medical history. Overly strict food restrictions can negatively impact a patient’s quality of life, so we advocate for minimizing these restrictions whenever possible, balancing the need for caution with the goal of maintaining a patient’s well-being [39].

In order to protect the health of consumers, the European legislator has required food operators who intend to serve raw or undercooked fish products to freeze them, with the right time/temperature ratio, which can kill larval forms if present [40]. Testing the molecular targets involved in allergen sensitization can be helpful in identifying patients sensitized to thermostable molecules with high allergenic potency. Unfortunately, only one non-species-specific molecule, tropomyosin, is commercially available using a single plex method, while for multiplex platforms only two specific *Anisakis* molecules are available. Furthermore, in the scientific literature, studies on *Anisakis* molecules conducted using ELISA are reported, but, using either home-made or commercial methods, these are labelled for research use only [34,41].

Therefore, to date, molecular assessment has shown the limitations in assisting patients with their dietary choices [8].

In summary, our findings have allowed us to identify a possible risk profile associated with *Anisakis*-related allergies. This involves an elevated likelihood of having encountered the parasite, which correlates with advancing age, likely attributed to increased exposure to raw/marinated fish over lifetime, and/or reporting dietary restrictions to the consumption of undercooked or raw fish.

The inability to validate the statistically significant association between BAT positivity and Ascaris- and tropomyosin-specific IgE positivity through multiple analyses can be elucidated by cross-reactivity phenomena among proteins that serological tests, being less specific, fail to resolve. The BAT second-level test, however, addressed this issue by excluding confounding factors through multiple logistic analysis, emphasizing its superiority in detecting such nuances [8,9]. Hence, while IgE serves as a recommended initial investigation in individuals with a clinical history suggestive of potential *Anisakis* sensitization due to its simplicity and cost-effectiveness, the application of the BAT is reserved for patients exhibiting a high likelihood of parasite sensitization. The non-automated nature and relatively higher cost associated with the BAT suggest that it should not be the primary diagnostic test for every suspected case of *Anisakis* allergy. This underscores the importance of employing a comprehensive diagnostic algorithm that includes skin prick tests (SPTs), *Anisakis*-specific IgE serology, and finally, the BAT, particularly for cases with elevated IgE levels [9]. This observation is noteworthy given the fact that the double-blind placebo-controlled food test (DPFCC), which is considered the gold standard for diagnosing food allergies, is not applicable in *Anisakis* allergy diagnosis [9].

It is essential to acknowledge certain limitations of this study. Firstly, the observational nature of the cross-sectional study introduces inherent limitations, coupled with potential concerns about representativeness due to the convenience sample and a limited pool of participants. Notably, variations in *Anisakis* positivity prevalence across different geographical areas linked to dietary habits should be considered, indicating that the results obtained in our sample may not be reflective of the general population [42,43]. Furthermore, it is important to note that our sample comprises a selection of potentially allergic individuals referred to allergy ambulatories, exposing our results to potential selection bias. Consequently, our findings should be validated through additional studies to be conducted on huge samples from the general population.

Lastly, the logistic regression allowed for the best balance of clarity, reliability, and appropriateness for small- to medium-sized datasets, such as this study. Despite more sophisticated recent approaches, in our opinion logistic regression remained preferable due to its fewer assumptions (e.g., linear discriminant analysis assumes normality and equal covariance matrices), more interpretable findings (compared with support vector machines or random forest classification), and less-intensive computational efforts without significant advantages (e.g., Bayesian logistic regression).

It is worth noting that the use of *Anisakis*-specific IgE, when using the analytical cut-off, seems to be well-suited for seroprevalence studies targeting at-risk populations. In contrast, the BAT, due to its execution method which relies on the analysis of freshly drawn blood samples that cannot be stored for an extended period, and for reasons of cost, is not ideal for large-scale epidemiological investigations. Our results, while confirming the excellent specificity of the BAT in detecting *Anisakis* allergy, as endorsed by the proposed comprehensive diagnostic algorithm for *Anisakis* allergy, advocate for the extensive use of *Anisakis*-specific IgE in population-based risk profiling [9].

In conclusion, in a context where there is growing interest in understanding *Anisakis*-related diseases, as documented by a number of relevant publications which has significantly increased over time, improving test sensitivity through analytical cut-off becomes essential [44]. A more sensitive test is not only useful for diagnostic purposes, enabling more accurate and timely diagnoses, but it may also have important epidemiological value, contributing to better understanding and control of the spread and impact of diseases associated with this parasite [18]. The proposed approach could potentially contribute to enhancing knowledge for stratifying populations based on health risks associated with *Anisakis* exposure in epidemiological settings characterized by an elevated consumption of marinated or raw fish, a recognized risk factor for *Anisakis* sensitization [20,45].

## Figures and Tables

**Figure 1 foods-13-02821-f001:**
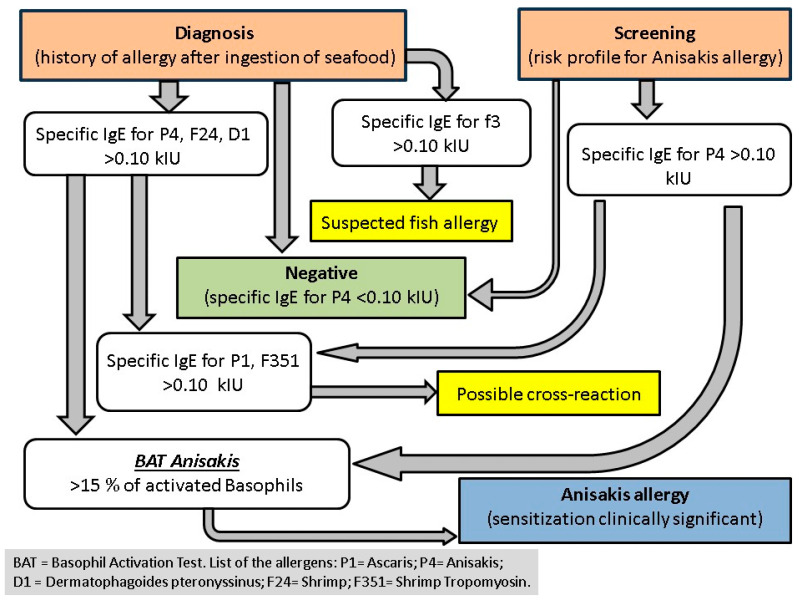
Flow chart for the diagnosis and the screening of *Anisakis* allergy.

**Table 1 foods-13-02821-t001:** Socio-demographic characteristics of the 259 participants recruited in the study.

	Total	Males	Females	*p*-Value
**N. (%)**	259 (100%)	88 (34.0%)	171 (66.0%)	-
**Age, mean (SD) in years**	40.8 (20.5)	38.8 (21.0)	41.8 (20.2)	0.25
**Education**				
No qualification	11 (4.2%)	5 (5.7%)	6 (3.5%)	0.87
Primary school	32 (12.4%)	10 (11.4%)	22 (12.9%)	
Middle school	94 (36.3%)	34 (38.6%)	60 (35.1%)	
Secondary school	95 (36.7%)	31 (35.2%)	64 (37.4%)	
University degree	27 (10.4%)	8 (9.1%)	19 (11.1%)	
**Residence**				
Inland area	47 (18.1%)	14 (15.9%)	33 (19.3%)	0.52
Coastal rural area	80 (30.9%)	31 (35.2%)	49 (28.7%)	
Coastal urban area	132 (51.0%)	43 (48.9%)	89 (52.0%)	
**Years in the current residence**				
Less than 10 years	26 (10.0%)	5 (5.7%)	21 (12.3%)	0.09
More than 10 years	233 (90.0%)	83 (94.3%)	150 (87.7%)	

**Table 2 foods-13-02821-t002:** Distribution of specific IgE-positive patients by analytical and clinical cut-offs for *Anisakis* according to BAT positivity (≥15%).

Anisakis IgE	Total n. (%) n. 259 (100)	BAT Negative n. (%) n. 247 (95.4)	BAT Positive n. (%) n. 12 (4.6)	*p*-Value
**Analytical cut-off**				
Negative	211 (81.5)	211(85.4)	0 (0.0)	Ref.
Positive (>0.1 kUA/L)	48 (18.5)	36 (14.6)	12 (100.0)	<0.001
**Clinical cut-off**				
Negative	240 (92.7)	237 (96.0)	2 (16.7)	Ref.
Positive (>0.35 kUA/L)	19 (7.3)	10 (4.0)	10 (83.3)	<0.001

**Table 3 foods-13-02821-t003:** Univariate analysis of the variables in study, using the *Anisakis*-specific IgE analytical cut-off and the BAT levels.

	Total n. (%) 259 (100)	*Anisakis* IgE Negative (kUA/L <0.1) n. (%) 211 (81.5)	*Anisakis* IgE Positive (kUA/L ≥0.1) n. (%) 48 (18.5)	OR [IC95%]	*p*-Value	BAT Negative (<15%) n. (%) 247 (95.4)	BAT Positive (≥15%) n. (%) 12 (4.6)	OR [IC95%]	*p*-Value
**Mean age (SD)**	40.8 (20.5)	39.4 (19.6)	46.6 (23.2)	1.02 [1.01; 1.03]	0.03	40.1 (20.4)	54.3 (19.2)	1.04 [1.00; 1.07]	0.02
**Sex**									
**Male**	88 (34.0)	61 (28.9)	27 (56.2)	Ref.	Ref.	81 (32.8)	7 (58.3)	Ref.	Ref.
**Female**	171 (66.0)	150 (71.1)	21 (43.8)	0.32 [0.17; 0.61]	<0.001	166 (67.2)	5 (41.7)	0.35 [0.10; 1.17]	0.08
**Consumption of raw, marinated, or smoked fish**									
**No**	12 (4.6)	5 (2.4)	7 (14.6)	Ref.	Ref.	9 (3.6)	3 (25.0)	Ref.	Ref.
**Yes**	247 (95.4)	206 (97.6)	41 (85.4)	0.14 [0.04; 0.49]	0.002	238 (96.4)	9 (75.0)	0.11 [0.03; 0.61]	0.01
**Skin contact with fish ***									
**No**	18 (7.11)	10 (4.9)	8 (16.7)	Ref.	Ref.	15 (6.2)	3 (25.0)	Ref.	Ref.
**Yes**	235 (92.9)	195 (95.1)	40 (83.3)	0.16 [0.09; 0.72]	0.01	226 (93.8)	9 (75.0)	0.20 [0.05; 0.99]	0.04

SD = standard deviation; OR = odds ratio; * data missing for 6 outpatients.

**Table 4 foods-13-02821-t004:** Comparison between *Anisakis*, Ascaris, and tropomyosin positivity (specific IgE analytical cut-off) and the BAT levels: univariate analysis.

	Total n. (%) n. 259 (100)	*Anisakis* IgE Negative (kUA/L <0.1) n. (%) n. 211 (81.5)	*Anisakis* IgE Positive (kUA/L ≥0.1) n. (%) n. 48 (18.5)	OR [IC95%]	*p*-Value	BAT Negative (<15%) n. (%) n. 247 (95.4)	BAT Positive (≥15%) n. (%) n. 12 (4.6)	OR [IC95%]	*p*-Value
**Anisakis (kUA/L) ≥ 0.1**									
**Positive**	48 (18.5)	-				36 (14.6)	12 (100.0)	144.9 * [17.2; ∞]	<0.001
**Ascaris (kUA/L) ≥ 0.1**									
**Negative**	222 (85.7)	211 (100.0)	11 (22.9)	Ref.	Ref.	220 (89.1)	2 (16.7)	Ref.	Ref.
**Positive**	37 (14.3)	0 (0.0)	37 (77.1)	1379.3 * [166.5; ∞]	<0.001	27 (10.9)	10 (83.3)	37.5 [9.1; 277.0]	<0.001
**Tropomyosin (kUA/L) ≥ 0.1**									
**Negative**	249 (96.1)	211 (100.0)	38 (79.2)	Ref.	Ref.	241 (97.6)	8 (66.7)	Ref.	Ref.
**Positive**	10 (3.9)	0 (0.0)	10 (20.8)	115.4 * [12.7; ∞]	<0.001	6 (2.4)	4 (33.3)	19.5 [4.11; 86.2]	<0.001

SD = standard deviation; OR = odds ratio; * calculated by Haldane–Anscombe correction.

**Table 5 foods-13-02821-t005:** Multiple logistic regression model considering Anisakis-specific IgE (Anisakis-IgE) positivity (≥0.1) as the dependent variable.

Anisakis-IgE ≥ 0.1	Adj-OR *	95%CI **	*p*-Value
**Age**	1.07	1.02–1.11	0.002
**Sex**	0.04	0.01–0.27	0.001

* Adj-OR: odds ratio adjusted by IgE Ascaris ≥ 0.1 and IgE tropomyosin ≥ 0.1. ** 95%CI: 95% confidence interval.

**Table 6 foods-13-02821-t006:** Multiple logistic regression model considering BAT positivity (≥15%) as the dependent variable.

BAT ≥ 15%	Adj-OR *	95%CI **	*p*-Value
**Age**	1.09	1.02–1.16	0.008
**Angioedema**	0.01	0.00–0.69	0.030

* Adj-OR: odds ratio adjusted by IgE Ascaris ≥ 0.1 and IgE tropomyosin ≥ 0.1. ** 95%CI: 95% confidence interval.

## Data Availability

The data presented in this study are available on request from the corresponding author due to privacy reasons.

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
