# Peer review of "Screening of Anisakis-Related Allergies and Associated Factors in a Mediterranean Community Characterized by High Seafood Consumption"

_foods, 2024, doi:10.3390/foods13172821_

Round 1

Reviewer 1 Report

Comments and Suggestions for Authors

 I recommend adding justifications for the choice of logistic regression over other potential methods, such as Linear Discriminant Analysis, Support Vector Machines (SVM) for Classification, Random Forest Classification, and Bayesian Logistic Regression. Highlighting the reasons for selecting logistic regression—such as its interpretability, robustness, and suitability for small to medium-sized datasets—will strengthen the methodological rigor of the study. Additionally, consider addressing any potential limitations of the alternative methods and why they may not have been the best fit for this particular analysis.

Author Response

Comments and Suggestions for Authors

I recommend adding justifications for the choice of logistic regression over other potential methods, such as Linear Discriminant Analysis, Support Vector Machines (SVM) for Classification, Random Forest Classification, and Bayesian Logistic Regression. Highlighting the reasons for selecting logistic regression—such as its interpretability, robustness, and suitability for small to medium-sized datasets—will strengthen the methodological rigor of the study. Additionally, consider addressing any potential limitations of the alternative methods and why they may not have been the best fit for this particular analysis.

R: We are grateful to the reviewer for the suggestions. First, we would like to share the reasons that made us choice the logistic regression:

  • interpretability: logistic regression provides clear and interpretable results, making it easier to understand the relationship between the dependent and independent variables.
  • robustness: it is a robust method that performs well even with small to medium-sized datasets, which is the case in our study.
  • suitability: logistic regression is well-suited for binary outcome variables, which aligns with our research objectives.

On our opinion, despite alternative methods, such as Linear Discriminant Analysis, Support Vector Machines (SVM) for Classification, Random Forest Classification, and Bayesian Logistic Regression, have their merits, they were not selected due to the following reasons:

  • the Linear Discriminant Analysis assumes normality and equal covariance matrices, which may not hold in our dataset;
  • the Support Vector Machines (SVM) can be complex and less interpretable, making it harder to communicate findings;
  • the Random Forest Classification, while powerful, can be prone to overfitting and less interpretable;
  • the Bayesian Logistic Regression may not offer significant advantages over traditional logistic regression for our dataset size.

For the abovementioned reasons, we believe that logistic regression offered the best balance of interpretability, robustness, and suitability for our study’s needs. To better address our choice, we have updated the text as follows:

  • [section “Materials and Methods”, Page 3, lines 122-125] “The variables included in the model were selected based on their statistical significance in univariate analyses and their clinical relevance. The final multiple logistic model was evaluated for goodness-of-fit using the Hosmer-Lemeshow test, and multi-collinearity was assessed using the generalized Variance Inflation Factor (VIF).
  • [section “Results”, Page 7, lines 276-277] “the final model was well-fitted, with a Hosmer and Lemeshow goodness of fit test p-value of 0.941. The generalized variance-inflation factors ranged from 1.16 to 3.81, indicating low multicollinearity.”
  • [section “Discussion”, Pages 10 and 11, lines 552-558] “Lastly, the logistic regression allowed the best balance of clarity, reliability, and appropriateness for small to medium-sized datasets, such as this study. Despite more sophisticated recent approaches, on our opinion logistic regression remained preferable due to its fewer assumptions (e.g., Linear Discriminant Analysis assumes normality and equal covariance matrices), more interpretable findings (compared with Support Vector Machines or Random Forest Classification), and less intensive computational efforts without significant advantages (e.g., Bayesian Logistic Regression).

Reviewer 2 Report

Comments and Suggestions for Authors

This work is too preliminary. Only four Tables are provided in this manuscript. Is the statistical result meaningful? No model is used and testified in this research. The authors should well design this work, and the statistical data and results can be tested by Empirical model. Or, related model can be fitted, and the accuracy can be testified. In addition, if the examples are enough to evaluate this topic? Reliable statistical analysis need be conducted, and in-depth discussion should be made based on their findings and related references. Overall, the quality of this manuscript is poor. Reject is suggested.

Author Response

Comments and Suggestions for Authors

This work is too preliminary. Only four Tables are provided in this manuscript. Is the statistical result meaningful? No model is used and testified in this research. The authors should well design this work, and the statistical data and results can be tested by Empirical model. Or, related model can be fitted, and the accuracy can be testified.

In addition, if the examples are enough to evaluate this topic? Reliable statistical analysis need be conducted, and in-depth discussion should be made based on their findings and related references. Overall, the quality of this manuscript is poor. Reject is suggested.

R: We thank the reviewer for the valuable feedback provided. We sincerely appreciate the insights and the suggestions for improving our manuscript. We have acknowledged that our study provides preliminary findings. However, we believe that it provides a deeper understanding of Anisakis-related allergies in a Mediterranean community with high seafood consumption, and, therefore, this initial exploration could be useful for future and more comprehensive studies. We understand the importance of a more robust statistical analysis. Our study employed Fisher’s exact test and logistic regression to analyze data and we believe our statistical methods are appropriate for the study’s scope. Although the final multiple logistic model was not presented as a table or figure, the most important findings were reported in the text. Our sample size was limited, but it was sufficient to identify significant trends and associations. Anyway, we added further specifications and justifications for the use of the analytical models employed.

As we believe that logistic regression offered the best balance of interpretability, robustness, and suitability, for our study’s needs, to better address our choice we have updated the text as follows:

  • [section “Materials and Methods”, Page 3, lines 122-125] “The variables included in the model were selected based on their statistical significance in univariate analyses and their clinical relevance. The final multiple logistic model was evaluated for goodness-of-fit using the Hosmer-Lemeshow test, and multi-collinearity was assessed using the generalized Variance Inflation Factor (VIF).
  • [section “Results”, Page 7, lines 276-277] “the final model was well-fitted, with a Hosmer and Lemeshow goodness of fit test p-value of 0.941. The generalized variance-inflation factors ranged from 1.16 to 3.81, indicating low multicollinearity.”
  • [section “Discussion”, Pages 10 and 11, lines 552-558] “Lastly, the logistic regression allowed the best balance of clarity, reliability, and appropriateness for small to medium-sized datasets, such as this study. Despite more sophisticated recent approaches, on our opinion logistic regression remained preferable due to its fewer assumptions (e.g., Linear Discriminant Analysis assumes normality and equal covariance matrices), more interpretable findings (compared with Support Vector Machines or Random Forest Classification), and less intensive computational efforts without significant advantages (e.g., Bayesian Logistic Regression).

Lastly, more in-depth discussion has been provided [section “Discussion”, Pages 9 and 10, lines 383-520]: “Fish is strongly affected by the environment in which it lives and is therefore susceptible to be attacked by parasites. Some may pose a health risk to consumers if the fish is consumed raw or not fully cooked. In contrast, cooking or freezing is an effective way to ensure their safety. The presence of these parasites, in particular Anisakis, can be considered as a natural condition and not a sign of alteration itself. It is indeed possible that dead Anisakis larvae may contaminate raw food leading to sensitisation, but the procedures of cooking or freezing is considered to reduce the risk. As explained in literature, oral intake of seafood containing allergenic Anisakis proteins from dead larvae rarely results in sufficient blood allergen content to induce systemic symptoms of Anisakis allergy. These findings indicate that it is generally unnecessary to uniformly eliminate cooked or processed seafood from the diets of most patients. However, it is important to recognize that the ingestion of dead Anisakis larvae, even in cooked or processed forms, can still trigger allergic reactions in patients who are highly sensitive. Therefore, while managing dietary restrictions, it is essential to tailor recommendations to the individual sensitivities and medical history. Overly strict food restrictions can negatively impact a patient's quality of life, so we advocate for minimizing these restrictions whenever possible, balancing the need for caution with the goal of maintaining a patient's well-being. In order to protect the health of consumers, the European legislator has required food operators who intend to serve raw or undercooked fish products to freeze them, with the right time/temperature ratio, which can kill larval forms if present. Testing the molecular targets involved in allergen sensitization can be helpful in identifying patients sensitized to thermostable molecules with high allergenic potency. Unfortunately, only one non-species-specific molecule, such as tropomyosin, is commercially available using a singleplex method, while regarding multiplex platforms only two specific Anisakis molecules are available. Therefore, to date, the molecular assessment has showed limitations in assisting patients with their dietary choices.”

We hope that these revisions have improved the quality of our manuscript and, that at the same time, they will overcome your concerns.

Round 2

Reviewer 2 Report

Comments and Suggestions for Authors

Although the authors have the attitute to revise this manuscript, its quality is still need to improve.

Some comments:

1. Some old references need be replaced by the recent three years.

2. Ony four simple Tables are given in this manuscript. I think that additional 2-3 figures (or schemes) involving this topic, summarizing the main results and summarizing Anisakis-related allergies and associated factors need be provided. This quality of this manuscript need be further improved.

3. The advantage and disadvantage of this work need be well summarized, and the prospect of this subsject need be well described. What will be done in the future?

Author Response

- Although the authors have the attitude to revise this manuscript, its quality is still need to improve.

R.: We thank the reviewer for giving us the opportunity to improve the quality of the manuscript. Here follows a point-by-point answer to the comments.

  1. Some old references need be replaced by the recent three years.

R.: We have updated the references as suggested by the reviewer (see reference list n. 15,16,17,18,26,27, 42,45).

  1. Ony four simple Tables are given in this manuscript. I think that additional 2-3 figures (or schemes) involving this topic, summarizing the main results and summarizing Anisakis-related allergies and associated factors need be provided. This quality of this manuscript need be further improved.

R.: Following the suggestion coming from the reviewer, we have inserted a table reporting the results of the multiple logistic regression model (Page 7, from line 205) and we further provided an additional figure, representing a Flow Chart for the Diagnosis and the Screening of Anisakis Allergy, that summarizes the practical implications of the study findings. Consequently, we have implanted the conclusion section as follows (Page 7, lines 207-208): “In Figure 1 is showed a flow chart translating a summary of the previous results into practical implications for the diagnosis and the screening of Anisakis Allergy.”

  1. The advantage and disadvantage of this work need be well summarized, and the prospect of this subject need be well described. What will be done in the future?

R.: The “advantage and disadvantage” of the manuscripts have been better highlighted in the discussion and conclusion sections, as suggested by the reviewer. Please see Page 8, lines 232-234 (“In this scenario, the importance of using a highly sensitive test for diagnosing diseases induced by the Anisakis nematode is evident. The BAT appears to be particularly relevant as it allows for the verification of the clinical relevance of sensitization.”), Page 11, lines 343-345 (“Furthermore, in the scientific literature studies on Anisakis molecules conducted using ELISA are reported, but either home-made methods or, commercial, these are labelled for research use only.”), Page 11, lines 349-352  (“This involves an elevated likelihood of having encountered the parasite, which correlates with advancing age, likely attributed to increased exposure to raw/marinated fish over lifetime, and/or reporting dietary restrictions the consumption of undercooked or raw fish.”) and Page 12, lines 394-400 (“In conclusion, in a context where there is growing interest in understanding Anisakis-related diseases, as documented by the relevant number of publications, which has significantly increased over time, improving test sensitivity through analytical cut-off becomes essential. A more sensitive test is not only useful for diagnostic purposes, enabling more accurate and timely diagnoses, but it may also have an important epidemiological value, contributing to a better understanding and control of the spread and impact of diseases associated with this parasite.”).